# Two-step Uncertainty Network for Task-driven Sensor Placement

## Abstract

Optimal sensor placement achieves the minimal cost of sensors while obtaining the prespecified objectives. In this work, we propose a framework for sensor placement to maximize the information gain called Two-step Uncertainty Network(TUN). TUN encodes an arbitrary number of measurements, models the conditional distribution of high dimensional data, and estimates the task-specific information gain at un-observed locations. Experiments on the synthetic data show that TUN outperforms the random sampling strategy and Gaussian Process-based strategy consistently.

## 1 Introduction

Sensor placement is widely studied in the areas of environment monitoring (Hu et al., 2018; Nguyen et al., 2015), structural health monitoring(Ostachowicz et al., 2019) , security screening(Masoudi et al., 2016), and adaptive computed tomography(Ouadah et al., 2017). The optimal sensor placement maximizes the objectives with minimal cost of sensors. Given the model that maps each possible set of sensor locations to the objectives, the optimal sensor placement can be formulated as an optimization problem. However, the optimization is shown to be NP-hard(Garey & Johnson, 2002). Thus, approximate greedy algorithms of sequential sensor placement are proposed and then proved to be near optimal under the assumptions that the objectives are monotone and submodular(Nemhauser et al., 1978).

The diagram of a sequential sensor placement is shown in Figure 1a with the black arrows. The agent inquires at a feasible location to the physical model in each step and obtains the corresponding measurement. The obtained observations are used to make inference for specific tasks. For instance, in security screening tasks the observations are used to predict the distribution of the object's label. To make an accurate inference, the agent often optimizes the information gain in each step with respect to the feasible location. The corresponding objective is mutual information which is approved to give near optimal approximations in sequential sensing(Krause et al., 2008; Nemhauser et al., 1978).

To optimize the objective, a model that estimates the potential information gain at each possible location is necessary. The most generic method to model the unknown spatial phenomenon is Gaussian Process(GP) which incorporates the knowledge of observations and predicts the uncertainty at the un-observed locations. However, the Gaussian model assumption in GP does not perform well on high dimensional data such as images as generative models. In addition, GP inherently adapts the assumptions that the uncertainty at the un-observed locations is independent of the obtained measurements, making the GP based sequential sensing an open-loop control(Rasmussen, 2003). An alternative approach to this problem arises recently is Reinforcement Learning(RL)(Ha & Schmidhuber, 2018). However, the performance in RL is found to be of large variance and difficult to reproduce(Henderson et al., 2018).

In this work, we propose a framework for sensor placement to maximize the information gain called Two-step Uncertainty network (TUN). The pipeline of TUN is shown in Figure 1 with red arrows. TUN consists of two steps, namely the imagination step and the inspection step. TUN firstly "imagines" the possible measurements at the un-observed locations. Then it estimates the task-specific information gain with the imagined measurements along with the previous observations in the inspection step. Both steps are deployed with the pre-trained neural networks. Given the task-specific information gain at all the un-observed locations, the agent adapts a greedy algorithm to select the

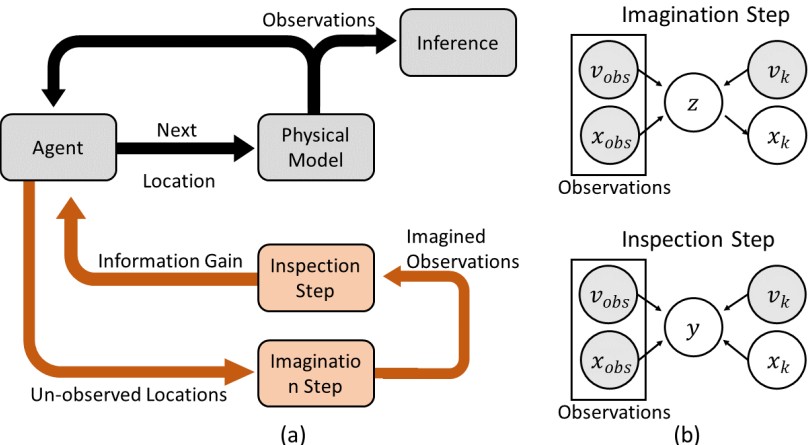

Figure 1: (a) The diagram of sensor placement (black arrows) and TUN(red arrows). TUN consists of two steps: imagination step and inspection step to generate the possible measurements and evaluate the task-specific information gain at un-observed locations. (b) Graphical models of the two steps in TUN. Instances of measurement $x_k$ at un-observed locations are generated in the imagination step. Then those generated measurements are used to evaluate the information gain for the task in inspection step.

optimal next location to inquire. This procedure emulates how we human think in such tasks: given the observations, we firstly imagine the possible outcomes at un-observed locations, then inspect the information pertaining to the task based on those possible outcomes. We will derive the proposed framework in the next section.

## 2 TWO-STEPS UNCERTAINTY NETWORKS

Consider a sequential sensing strategy, we denote locations as $v$ and measurements as $x$. At the $k^{th}$ step, we have the previous $k - 1$ observations $Obs = \{x_1, v_1, ..., x_{k-1}, v_{k-1}\}$, the optimal location $v_k^*$ is the one maximizes the mutual information (MI) between the object's label $y$ and the possible measurement $x_k$ given the previous observations:

$$v_k^* = \arg\max_{v_k} MI(y; x_k | Obs, v_k) \tag{1}$$

The mutual information can be expressed as,

$$MI(y, x_k | v_k, Obs) = H(y | Obs) - \mathbb{E}_{Pr(x_k | v_k, Obs)} H(y | x_k, v_k, Obs) \tag{2}$$

where $\mathbb{E}$ is the expectation operator. In Equation 2, the first term is the uncertainty of labels conditioned on the previous observations. The second term is the expected uncertainty conditioned on observations and possible measurements $x_k$ at $v_k$. The subtraction gives the uncertainty reduction or the information gain at the location $v_k$. It is worth noting that the first term is independent of $v_k$, and can be treated as a constant in optimizing Equation 2 with respect to $v_k$. The second term in Equation 2 can be approximated with Monte Carlo estimator as

$$MI(y, x_k | Obs) = - \sum_{m=1}^{M} \frac{1}{M} H(y | x_k^m, v_k, Obs) + Const. \tag{3}$$

where

$$x_k^m \sim Pr(x_k | v_k, Obs) \tag{4}$$

The summation in Equation 3 (without the negative sign) is the approximate remaining entropy with the measurement at $v_k$ given. Maximizing the mutual information is equivalent to minimizing the remaining entropy. Equation 4 is the conditional distribution of the measurement at $v_k$.

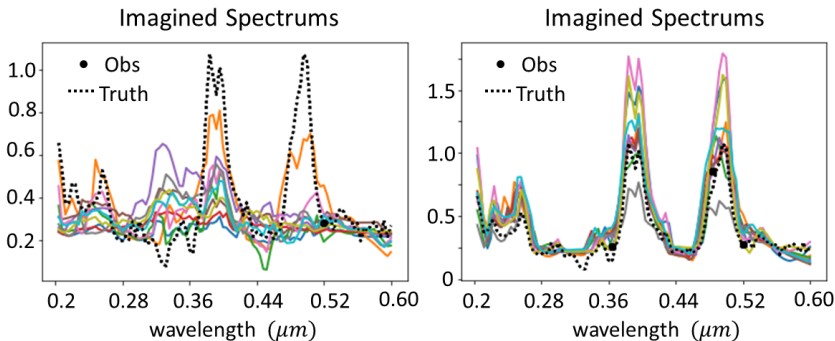

Figure 2: Example of the generation step on 1D spectrum dataset. The dashed line is the true spectrum and the solid spots are the observations. The generated instances are in colorful solid lines. Left: 10 imagined spectrums based on single observation. Right: 10 imagined spectrums based on three observations. The variation in the generated samples is mainly from scales and is independent of the task.

Following Equation 3-4, it is natural to approach the remaining entropy in two steps : (1), Generating instances of $x_k$ that follows the distribution in Equation 4. (2), Evaluating the remaining entropy $\sum \frac{1}{M} H(y|x_k^m, v_k, Obs)$ with the generated instances in step (1). The graphical models of the two steps are shown in Figure 1b. The design of our Two-step uncertainty network (TUN) follows the same rationale. In the imagination step, TUN generates multiple instances with a generative neural network. In the inspection step, a deterministic deep neural network is used to estimates the label distribution, and thus to evaluate the information gain(or negative remaining entropy).

## 2.1 IMAGINATION STEP

The first step of TUN is to generate instances $x_k \sim Pr(x_k|v_k, Obs)$. Modeling the distribution of high dimensional data such as images is difficult and computationally expensive, even within a constrained family of distributions with simplified assumptions. Recently, remarkable progress has been made on modeling complex distribution of high dimensional data with generative neural network(GNN)(Gregor et al., 2015; Eslami et al., 2018). GNN maps the instances of re-parameterizable distribution (for example, multivariate normal distributions) to the instances of target complex distribution(Kingma & Welling, 2013). GNN is trained to maximize the log likelihood of the generated instances,

$$\log Pr(x_k|v_k, Obs) = \log \int Pr(x_k|z, v_k) Pr(z|Obs) dz \qquad (5)$$

where $z$ is the latent variable of multivariate normal distributions. The log likelihood within the integral is intractable. Thus, the evidence lower bond(ELBO) as an approximation is evaluated instead,

$$\ln \int Pr(x_k|z, v_k) Pr(z|Obs) dz \geq \mathbb{E}_{q_\phi(z)} \ln Pr(x_k|z, v_k) - D_{KL}[q_\phi(z); p_\theta(z)] \qquad (6)$$

The posterior distribution $q_\phi(z|x_k, v_k, Obs)$ is conditioned on the previous $k-1$ observations $Obs$ and the observed $k^{th}$ measurement in the training data. The prior distribution $p_\theta(z|v_k, Obs)$ is only conditioned on $Obs$. $D_{KL}$ is the KL divergence measuring the difference between the two distributions. After training, we will have a generator network $G(v_k, Obs)$ that generates instances of measurement $x_k$. The generator network $G$ is then employed in the imagination step of TUN.

## 2.2 INSPECTION STEP

The second step in TUN is inspection, which estimates the task-specific information gain(or negative remaining entropy) at $v_k$. With the generated $M$ instances of $x_k$ in the imagination step, the

remaining entropy in Equation 3 can be expressed as

$$\sum_{m=1}^{M} \frac{1}{M} H(y|x_k, v_k, Obs) = -\sum_{m=1}^{M} \frac{1}{M} \sum_{y} Pr(y|x_k^m, v_k, Obs) \log Pr(y|x_k^m, v_k, Obs) \quad (7)$$

The remaining entropy is a function of the conditional distribution of the label $Pr(y|x_k^m, v_k, Obs)$. We approximate this conditional distribution with a deterministic neural network, the inspector network $D(x_k, v_k, Obs)$. Then the remaining entropy is evaluated as Equation 7. The most informative location to inquire is the one with the lowest remaining entropy. The procedure is summarized in algorithm 1. It can be shown that the true parameters of the model can be recovered by symptomatically maximizing a proper scoring rule(Gneiting & Raftery, 2007). A proper scoring rule $S$ rewards the true distribution $p$ more than any other distributions $\hat{p}$ on the training data $d$ as

$$\int_d p(d)S(\hat{p}, d) \leq \int_d p(d)S(p, d) \quad (8)$$

It is shown that optimizing the softmax cross entropy loss function in the case of multi-class classification is equivalent to optimizing a proper scoring rule(Lakshminarayanan et al., 2017). Thus, the inspector network $D$ is trained with the softmax cross entropy loss. It is worth noting that the inspector network $D$ takes different number of observations at different steps. To accept arbitrary number of observations, each observation is encoded separately with a shared-weight encoder, and the encoded vectors are fed into an aggregator and aggregated to a fixed-length vector before the succeeding networks. To enforce the commutative property in the sequential sensing problem, we adapt a "mean operator" as the aggregator in TUN, which takes the average of the input vectors. We take random number of observations at the randomly selected locations in the training stage.

---

**Algorithm 1** At the $k^{th}$ step of TUN

---

**Require:** $Obs, G, D, m = 1$
**Ensure:** The optimal $k^{th}$ location $v_k^*$
    **for** $j$ is in un-observed locations **do**
        $v_k = j$
        **for** $m \leq M$ **do**
            $x_j^m \sim G(Obs, v_k)$
            $P_y^m = D(Obs, v_k, x_j^m)$
            $H_y^m = -P_y^m \ln P_y^m$
            $m+=1$
        **end for**
        $H_j = \frac{1}{M} \sum_m H_y^m$
    **end for**
    **return** $v_k^* = \arg \max_j H_j$

---

## 3 EXPERIMENTS AND RESULTS

To evaluate the feasibility of TUN, we experimented with synthetic datasets. In the first experiment, we visualize the imagination step in TUN with simple 1D spectrum dataset. The spectrum dataset is generated from the spectrums of five minerals including Augite, Allanite, Xenotime, Bikitaitem and Pseudobrookite. We re-sampled the spectrums from $0.2\mu m$ to $0.6\mu m$ with 100 points and normalized them. The normalized spectrums are then scaled by a random factor ranging from 0.025 to 2.5 and corrupted by a zero-mean Gaussian noise with standard deviation of 0.03. The random scaling creates the intra-class uncertainty in the dataset. We prepared 5000 instances in the training dataset and 500 instances in the test dataset. The generator network $G$ was trained to generate the instances of the spectrum with several observations given. The number and locations of the observations are randomly selected in the training process. In the test stage, we show 10 generated instances in colorful solid lines in Figure 2 with different observations. The observations are indicated as filled circle, and the true spectrum is shown in the dashed line. Given the single observation, the generated instances vary in both mineral types(inter-class uncertainty) and scales(intra-class uncertainty). While with three selected observations, the imagined spectrums mainly vary in scales. The

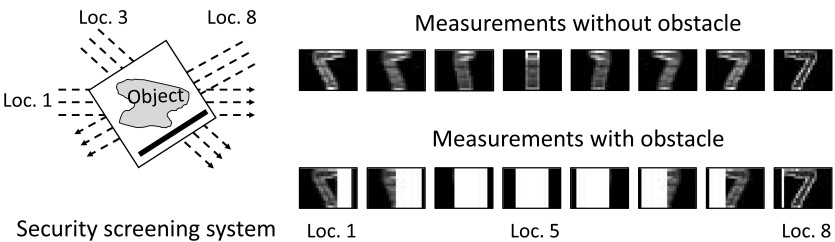

Figure 3: Left: x-ray security screening system. 8 different directions(locations) uniformly distributed within $180°$ are available to illuminate the object . The object is randomly rotated along $z$ axis with an obstacle. Additive Gaussian noise is adapted on the measurements. Right: examples of measurements at 8 locations.

intra-class uncertainty in the latter case is task-independent information indicating that the generator network $G$ believes not much information of label are remaining given the observations. In the case of that, taking more measurements may not benefit the label prediction much. Thus, the agent may stop inquiring to avoid redundant inquiries. We will show more quantitative results of the evaluation of the task-specific information gain in the inspection step of TUN in high dimensional dataset.

To quantitatively evaluate the information gain(or uncertainty reduction), we created a high dimensional synthetic dataset from x-ray baggage scanning system in security screening. The physical model is shown in Figure 3. The objects to be screened are 3D digits with an obstacle that partially blocks the objects. The existence of the obstacle results in significant variation of information among different locations. There are 8 locations(angles) to illuminate the x-ray onto the object (ranging from $0°$ to $157.5°$). Inquiry on each location returns a 2D projected image. The goal is to recognize the label of the object confidently with least number of inquiries. Before any inquiries, a randomized rotation is applied on the object along z axis. An additive Gaussian noise $n \sim (0, 0.02)$ is adopted on the observed images. TUN is trained on 5000 objects with random locations, effectively making the training dataset much larger. TUN is tested on 3000 set of observations generated from 1000 held-out un-seen objects. The generator network $G$ firstly encodes the information of one or more observations into a fixed length representation vector. Then it generates $M$ instances of possible measurements at the un-observed locations. based on the representation vector. In our model, $M = 10$. We show this process in Figure 4. In Figure 4 Left, the first measurement at location 4 is observed. We can see a corner feature in the measurement in the yellow box. Obviously the information from the observation is insufficient to reconstruct the 3D object not to mention the label of the object. We show three generated instances from the generator network at location 6 and 7 which are different in labels(digit 7 and 0) and fonts, yet consistent with the observation. In Figure 4 Middle, measurements at location 4 and 5 are both obtained. The generated samples at location 6 and 7 are more convergent to the ground truth as more information in observations are extracted. The generator network in this situation almost collapses to a deterministic neural network. This shows that our generator network generates the samples following the distribution $x_k \sim Pr(x_k|v_k, Obs)$.

In the second step of TUN, the generated samples are fed into the inspector network $D$, which estimates the probability of the labels $Pr(y|x_k, v_k, Obs)$ and evaluates the task specific information gain. We will perform both qualitative and quantitative analysis on the task specific information gain with TUN in the following paragraph. Firstly, we visualize the intermediate feature space in the initial sensing step of an example shown in Figure 5 Left. The obtained observations at location 5 is non-informative. We generated 100 instances at each location and fed them into the inspector network and visualized the feature space in the inspector network using t-SNE(Maaten & Hinton, 2008). We select the vector at the layer before logits in the inspector network to visualize. The feature space is colored by locations and divided into three regions. The region 1 covers the features of the generated instances from exact observed location (location 5). Region 2 covers the features from the locations close to the observed one (location 4 and 6). Region 3 contains the features from the locations far from the observed location. Clearly the features get more disperse as the distance to the observed location increases. This indicates that rich information lies in the locations in region 3. Although this is a qualitative analysis on the feature space, it justifies the necessity to

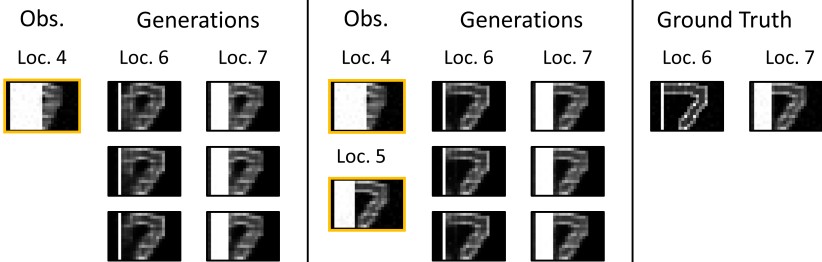

Figure 4: The imagination step of TUN on high dimensional dataset. Left: The 3 instances of the generations at location 6 and 7 given the observations at location 4. The generated instances are different in labels(digit 0 and 7) and fonts, but they all keep the corner feature occurred in the observation.Middle: The generations given two observations at location 4 and 5. The generated instances are much more convergent as more information is extracted from the observations. Right: The ground truth of the measurements at location 6 and 7.

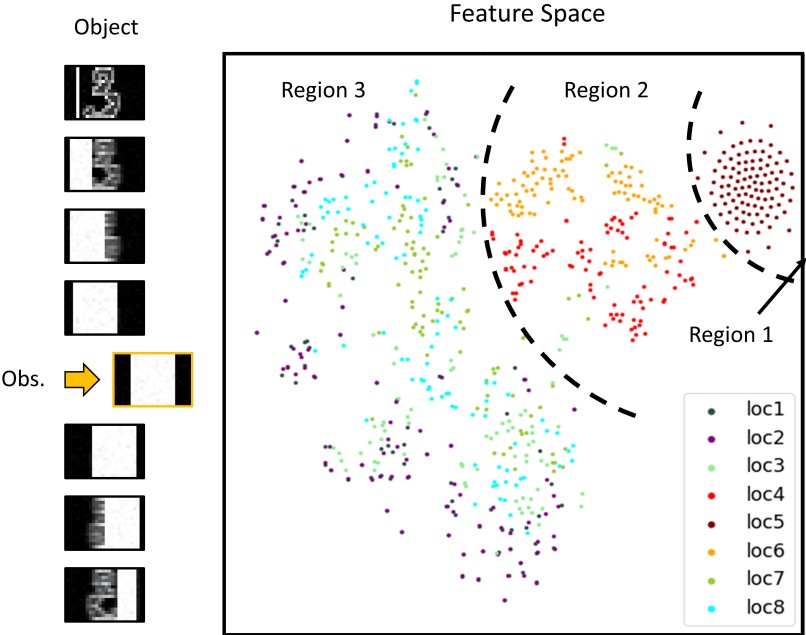

Figure 5: Task-specific feature distribution. We visualize the initial intermediate feature distribution in the inspector network using t-SNE. Clearly the features become more scatty at further locations from the observed location. The feature space is colored by locations. Region 1: Features of generated instances at the observed location. Region 2: Features at locations that are close to the observed location. Region 3: away from observed location. Region 3: Features at locations far from the observed locations.

sample multiple instances from the generator network. We will perform quantitative analysis of this example with the inspector network in next paragraph.

The information gain is evaluated from the averaged entropy in Equation 7 from $M$ samples. The averaged entropy indicates the estimated remaining uncertainty of the labels after we obtain the observation at location $v_k$. Our strategy is to pick the location with least remaining uncertainty, which equivalently maximizes the mutual information in Equation 2 (note the negative sign before the entropy). We show this quantitatively in the example in Figure 6. This is the same example as described in Figure 5. The first observation is at location 5 shown in yellow box in Figure 6a bottom, which is a non-informative observation. The averaged entropy for next step is shown in Figure 6a top. The entropy plot estimates the potential remaining uncertainty at the un-observed

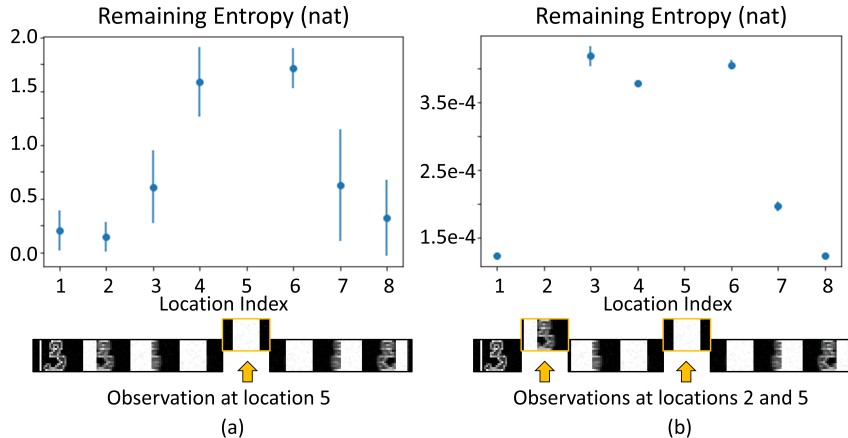

Figure 6: Entropy estimation: (a) The remaining entropy at un-observed locations given a single observation at location 5. The optimal next location in this step is location 2 at which the expected remaining entropy is least. (b) As we obtain the measurement at location 2 following TUN, the estimated remaining entropy at all the un-observed locations are almost zeros, indicating that the model is quite confident about the label with those observations.

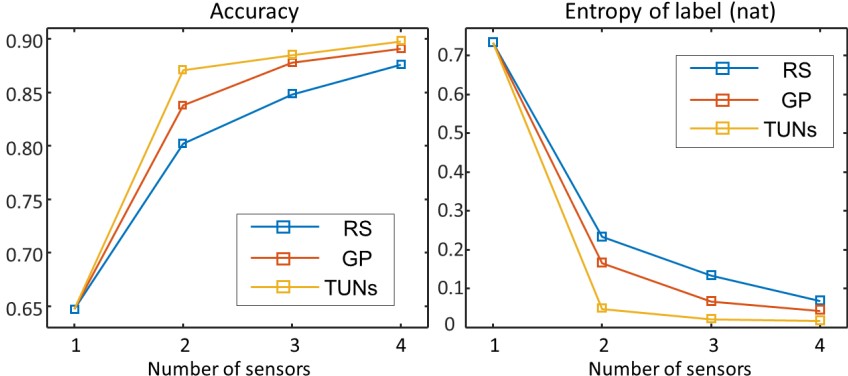

Figure 7: Accuracy and entropy at different numbers of inquiries for TUN, Gaussian process(GP) based and random sampling(RS) strategies. The advantages of sensor placement strategies starts at two observations. The proposed framework, TUN, outperforms GP and RS strategies consistently.

locations, thus the less remaining uncertainty the better the location is. The entropy is averaged from 10 samples, and the standard deviation is shown as bars in Figure 6. The entropy plot at the initial step indicates that our model believes there is less remaining uncertainty after we obtain the measurement at location 1,2, or 8 than that at location 3,4,6, or 7. Thus the next location selected by TUN is location 2 (which has least averaged remaining entropy). The successive estimation for the entropy is shown in Figure 6b in which the agent inquires and obtains the observation at location 2 following TUN. With the observations at location 2 and 5, the remaining entropy is very low with neglectable variance. This entropy plot shows TUN is quite confident on the label of the object, and believes there is not much information left at un-observed locations. A threshold can be used as a stopping criterion in practice.

We compare TUN with random sampling strategy(RS) and Gaussian Process(GP) strategy. We adapt squared exponential kernel in GP and employ the 2D coordinates and the projection angle $[x, y, cos\theta, sin\theta]$ as features. The GP model is fitted with training data and performs the prediction of measurement at un-observed locations. To evaluate the quality of the strategies, we trained classifiers with different number of the observations. The training data for the classifiers are generated from 5000 held-out objects with random noise and rotation. All the observations in training the classifiers are taken from random sampling strategy. The performance in both accuracy and

entropy(confidence) with different sensing budget are shown in Figure 7. The first location is randomly selected in all three strategies, thus, the performances are the same for all strategies. Start from the second step, TUN outperforms other strategies consistently with higher accuracy and less uncertainty.

## 4 CONCLUSION

In this work, we present a task-driven sensor placement framework to maximize the information gain. The proposed framework (TUN) is able to perceive and understand the observation, approximate the conditional distribution of the object, and estimate the information gain pertaining to the task. In the security screening experiment we demonstrated, TUN outperforms random strategy and GP strategy consistently.

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
