# OpenReview forum: "TWO-STEP UNCERTAINTY NETWORK FOR TASKDRIVEN SENSOR PLACEMENT"
_ICLR.cc/2020/Conference — Reject_

### Official Review · AnonReviewer1 · 2019-10-23
**Official Blind Review #1**

**Rating:** 1

**Review:**

This paper describes a sensor placement strategy based on information gain on an unknown quantity of interest, which already exists in the active learning literature. As is well-known in the literature, this is equivalent to minimizing the expected remaining entropy. What the authors have done differently is to consider the use of neural nets (as opposed to the widely-used Gaussian process) as the learning models in this sensor placement problem, specifically to (a) approximate the expectation using a set of samples generated from a generator neural net and to (b) estimate the probability term in the entropy by a deterministic/inspector neural net. The authors have performed some simple synthetic experiments to elucidate the behavior and performance of their proposed strategy.

Conventionally, the sensor placement strategy is tasked to gather the most informative observations (given a limited sensing budget) for maximimally improving the model(s) of choice (in the context of this paper, the neural networks) so as to maximize the information gain. The authors seem to have adopted a different paradigm in this paper: Large training datasets are needed for the prior training of both neural nets (in the order of thousands as reported in the experiments). This seems to be defeating the original aim/objective of sensor placement, as described above. Consequently, it is not clear to me whether their proposed strategy would be general enough for use in sensor placement for a wide variety of environmental monitoring applications. Random sampling and GP-based sensor placement strategies do not face such a severe practical limitation.

The paper is also missing several important technical details and clarity of presentation is poor. For example,

(a) The configurations and training procedure of generator NN G and deterministic NN D for the experiments are not sufficiently described for each experiment.

(b) What do the authors do with the new observations obtained from placing the sensors in the last experiment? Do they adopt an open-loop sensor placement strategy?

(c) The setup for the last experiment is not clear. Is it still the same object classification task? Is the GP receiving an exclusive set of 4D features that are different from the other two methods? I get the impression that the classifiers are trained a priori. For the GP classifier, isn't it the case that one should gather the most informative observations to maximally improve its classification accuracy?


Though I like the authors' motivation of the setup of the x-ray baggage scanning system in security screening, what has really been done in their experiments appears to be still quite far from this real-world setup. Furthermore, their proposed strategy has been used to gather only 1 to 4 observations. More extensive empirical evaluation with real-world datasets (inspired by realistic problem motivation) is necessary.

Fig. 2: I find it surprising that with a single observation, it is possible to generate the instance/imagined spectrum in orange that resembles that of the true spectrum. Similarly, with 3 observations, all 10 instances/imagined spectrums can exhibit the first spike (without observations on it). Can the authors explain this phenomena?



Minor issues
The authors need to put a space in front of all opening round bracket. Other formatting issues exist.
Equation 3: v_k is missing from the conditioned part on the lefthand side of the equation.
Page 3: to estimates?
Page 3: evidence lower bond?
Figure 2 appears on page 3 and is only referenced on page 4.
Algorithm 1: The use of subscript j in x^m_j to represent an unobserved location confuses with that of subscript k in x^m_k to denote the time step.
Figure 3 captions: adapted on the measurements?
Figure 4 captions: corner feature occurred?


**Experience Assessment:**

I have published in this field for several years.

**Review Assessment: Checking Correctness Of Derivations And Theory:**

I carefully checked the derivations and theory.

**Review Assessment: Checking Correctness Of Experiments:**

I carefully checked the experiments.

**Review Assessment: Thoroughness In Paper Reading:**

I read the paper thoroughly.

---

### Official Review · AnonReviewer3 · 2019-10-24
**Official Blind Review #3**

**Rating:** 1

**Review:**

Summary:
This paper addresses the issue of how to optimize sensor placement. The authors propose a framework for sensor placement called Two-step Uncertainty Network (TUN) based on the idea of information gain maximization. More concretely, the proposed method encodes an arbitrary number of measurements, models the conditional distribution of high dimensional data, and estimates the task-specific information gain at unobserved locations. Experimental results on the synthetic data clearly show that TUN outperforms current state-of-the-art methods, such as random sampling strategy and Gaussian Process-based strategy.

Comments:
- On page 1, the phrase “… on high dimensional data such as images as generative models” seems unclear.
- The lhs of Eq 3. should be MI(y,x_k|v_k, Obs)?
- In Fig. 1a, the “red arrows” for indicating TUM look like brown?
- In Fig. 1b, only the variable x_k is mentioned in the caption.
- If I understand correctly, on page 3 in Sect. 2.1, the imagination step is basically the same as VAE? After all, there does not seem to be any discussion on the choice of variational approximation q_{\phi} and prior p_{\theta}(z), where is crucial for performing variational inference.

Though I am not an expert in this domain, I find the basic idea is simple and easy to understand. However, my major concern is about the novelty of this work, given the fact that the theoretical contribution is quite limited.

**Experience Assessment:**

I have read many papers in this area.

**Review Assessment: Checking Correctness Of Derivations And Theory:**

I assessed the sensibility of the derivations and theory.

**Review Assessment: Checking Correctness Of Experiments:**

I assessed the sensibility of the experiments.

**Review Assessment: Thoroughness In Paper Reading:**

I read the paper at least twice and used my best judgement in assessing the paper.

---

### Decision · Program_Chairs · 2019-12-19

**Decision:**

Reject

**Comment:**

This paper proposes a sensor placement strategy based on maximising the information gain. Instead of using Gaussian process, the authors apply neural nets as function approximators. A limited empirical evaluation is performed to assess the performance of the proposed strategy.
The reviewers have raised several major issues, including the lack of novelty, clarity, and missing critical details in the exposition. The authors didn’t address any of the raised concerns in the rebuttal. I will hence recommend rejection of this paper.